

# Short-term wind speed prediction of wind farm based on TSO-VMD-BiLSTM

Qi Wang and Lei Zhang

School of Electrical and Information Engineering, Jiangsu University of Technology, Changzhou, Jiangsu Province, China

## ABSTRACT

Aiming at the random and intermittent characteristics of wind speed, a short-term wind speed prediction (SWSP) method based on TSO-VMD-BiLSTM is proposed in this article. Firstly, open-source historical data from a certain region in 2022, including wind speed, direction, pressure, and temperature is analyzed. The data is processed through variational mode decomposition (VMD) to fully extract feature data from historical wind speed records. Secondly, taking historical wind speed, direction, pressure, and temperature as inputs and wind speed as output, a SWSP model based on long short-term memory (LSTM) network is constructed. Thirdly, the tuna swarm optimization (TSO) algorithm is utilized for parameters optimization, and a bi-directional long short-term memory (BiLSTM) network is incorporated to enhance prediction accuracy for micrometeorological parameters. The proposed TSO-VMD-BiLSTM model is validated through comparison with other models, demonstrating its higher accuracy with the maximum absolute error of only 2.52 m/s, the maximum root mean square error of 0.81, the maximum mean absolute error of only 0.54, and the maximum mean absolute percentage error of 6.89%.

# INTRODUCTION

The penetration rate of wind energy as a renewable energy source in the power grid has significantly increased in recent years (*Jiang, Jia & Guan, 2019*). However, the randomness and intermittency of wind speed present challenges to stable power generation and grid integration, potentially disrupting the power system stability (*Sema, Başak & Gülbahar, 2022*). Consequently, precise wind speed prediction becomes crucial.

Extensive research has been undertaken on short-term wind speed prediction (SWSP) from diverse perspectives, leading to the categorization of wind speed prediction methodologies into three primary types: (1) The physical method, which involves collecting various local meteorological data, such as wind direction, temperature, pressure, and humidity, coupled with the use of geographical characteristics and meteorological information to construct hydrodynamics and thermodynamics models for predicting wind speed. Although straightforward in principle and relatively easy to implement, this method demands considerable computational resources. The model development and

Corresponding author
Qi Wang, wangqitz@163.com

solution processes are time-intensive, rendering it more suitable for medium to long-term predictions of wind speed (*Ai, Li & Xu, 2022*). (2) The statistical method, which depends heavily on extensive historical wind speed data, seeking to uncover underlying patterns through time series analysis of wind speed, hence its alternate name: the time series method. This approach typically necessitates a significant volume of random and stationary data for model construction, alongside accurate identification of model parameters. Compared to the physical method, the accuracy of the statistical method is improved, However, struggles to accurately model the nonlinear relationship among variables well (*Yang, Deng & Chang, 2022*). (3) The machine learning method (*Gupta, Natarajan & Berlin, 2022*), which utilizes the robust self-learning capabilities of machine learning algorithms to extract feature variables from the data, identifying potential nonlinear and strongly coupled relationships between input and output variables for wind speed prediction. Currently, this method represents the focal point of research in the field, noted for its high prediction accuracy and the flexibility of its algorithms. However, it usually requires integration with intelligent optimization algorithms to optimize the parameters of the prediction model.

In this article, a short-term wind speed prediction (SWSP) method based on tuna swam optimization-variational mode decomposition-bi-directional long short-term memory (TSO-VMD-BiLSTM) is proposed by combining the above three methods. This method employs VMD to decompose the wind speed time series and fully extract feature data. Based on historical data, a BiLSTM model for SWSP is established, and the secondary traversal of BiLSTM can capture the influence of weak meteorological parameters on wind speed prediction. To enhance the accuracy of SWSP, the BiLSTM prediction model by leveraging its advantages of fast convergence speed, high adaptability, and strong optimization ability offered by TSO.

The organizational structure of this article is arranged as follows. 'Acquisition and processing of short-term wind speed data' is the collection and processing of short-term wind speed data, and 'Basic principles of LSTM, BiLSTM and TSO' covers the basic principles of LSTM, BiLSTM, and TSO. In 'Results', the SWSP results based on TSO-VMD-BiLSTM in Example 1 and Example 2 are presented, and the results are fully discussed. The conclusion of this article is arranged in 'Conclusions'.

## ACQUISITION AND PROCESSING OF SHORT-TERM WIND SPEED DATA

### Wind speed data acquisition

This article takes the open-source wind speed data of a certain wind farm in 2022 as the research object, and draws the line chart of monthly average wind speed distribution, as shown in Fig. 1. As can be seen from Fig. 1, the average monthly wind speed in January is the lowest, the average monthly wind speed in May is the highest, and the average wind speed tends to stabilize in other months. To ensure the representativeness of SWSP, this article selects specific months to conduct example studies. Specifically, January, characterized by its lowest average monthly wind speed, is chosen as an example of a valley month. On the other hand, May, known for its highest average monthly wind speed, is selected as an

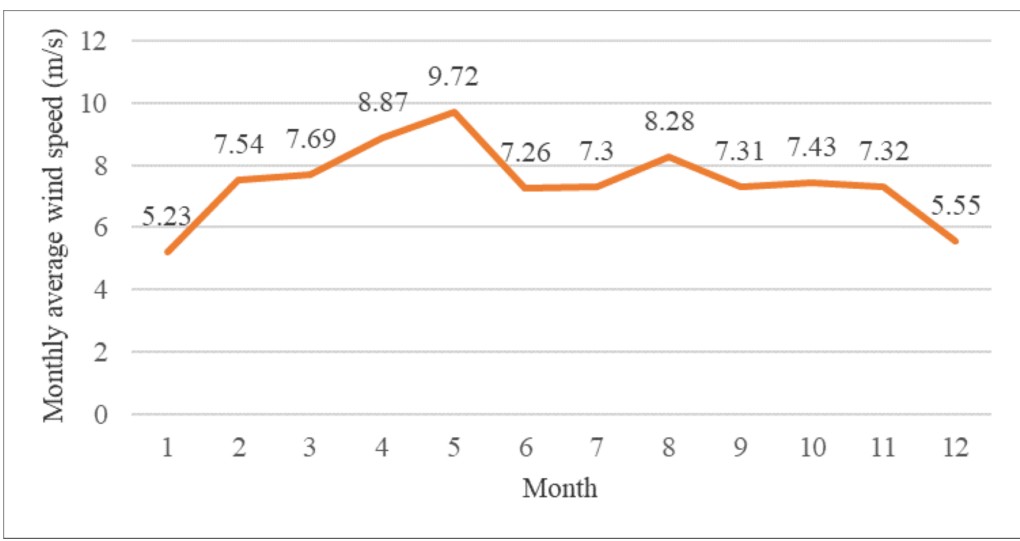

**Figure 1** Monthly average wind speed of a wind farm.

example of a peak month. This approach takes into account the randomness, variability, and periodicity of wind speed in order to provide a comprehensive analysis of SWSP. The data for Example 1 is from January 1 to January 31, 2022, and the data for Example 2 is from May 1 to May 31, 2022. The input variables of both data sets are wind speed, direction, pressure and temperature, and the output variable is wind speed.

Due to various factors such as natural and human factors that may affect the collection of meteorological data (*Dong et al., 2022*), occasional incomplete data may occur in the collected meteorological data. For a small amount of individual missing data, this article supplements it by using the mean of adjacent dates to ensure data continuity.

Figures 2 and 3 show the wind speed distribution series of the wind farm in January and May, respectively. It is apparent that the randomness and fluctuation of wind speed are more obvious in these two wind speed series, which directly leads to an increase in the difficulty of prediction. Both datasets consist of 2,976 sample sequences, and the training and testing sets are partitioned in an 8:2 ratio.

## Wind speed data processing

Wind speed data collected by wind tower in wind farm is usually affected by external environment or other uncontrollable factors, which may bring noise information into the data. Therefore, it is necessary to process the collected historical data in advance to improve the characteristics of the data.

VMD can decompose wind speed data of the same category and time period, further obtaining wind speed sub sequences with different frequencies but stronger regularity, thereby reducing the complexity of wind speed sequences (*Wang, Wei & Teng, 2023*). Each component decomposed by VMD has an independent center frequency, thus multiple sub sequences with different frequencies and corresponding features can be obtained.

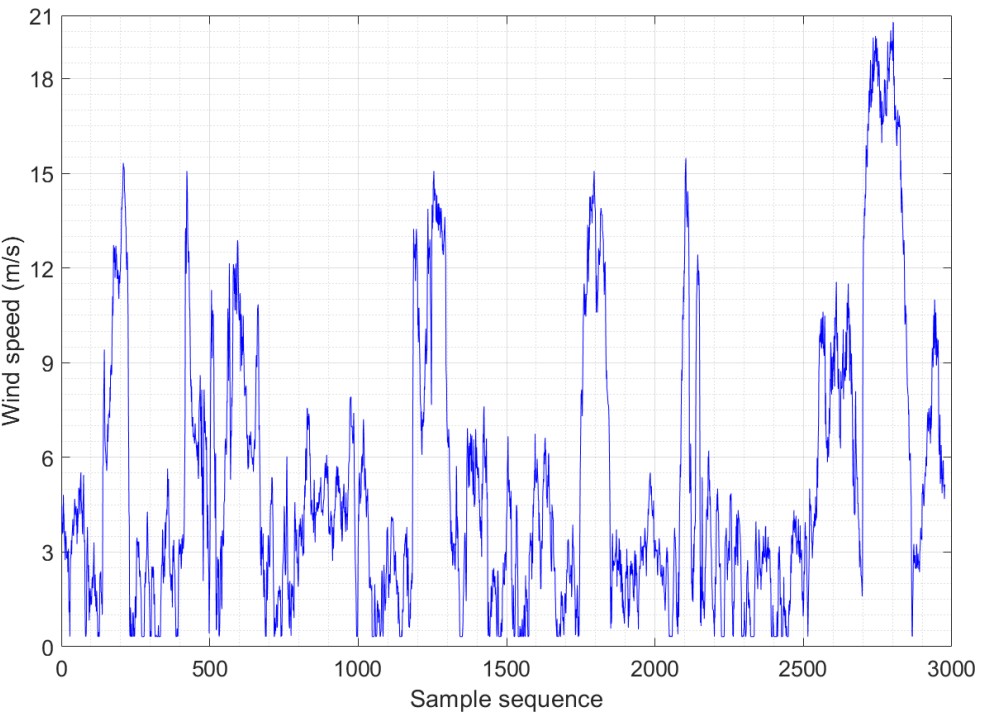

**Figure 2** Wind speed distribution sequence diagram of Example 1.

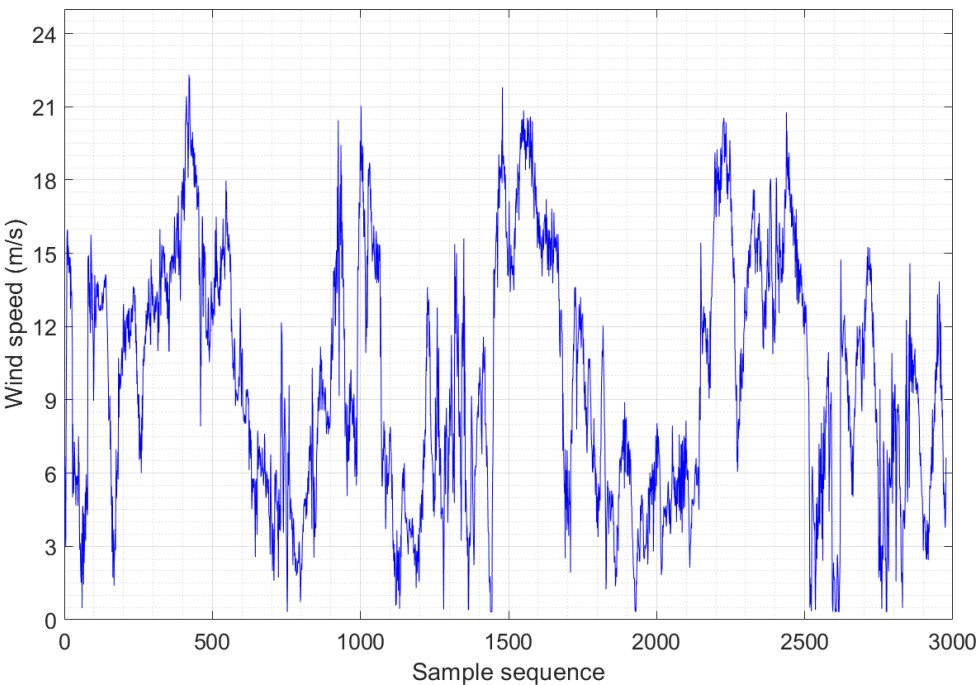

**Figure 3** Wind speed distribution sequence diagram of Example 2.

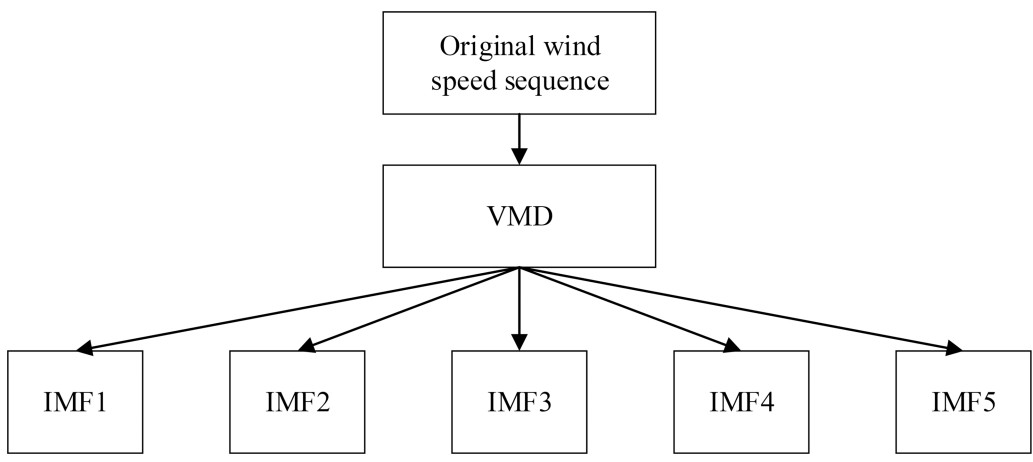

**Figure 4** **VMD processing process of wind speed data.**

In addition, VMD can handle both stationary and non-stationary sequences (*Ye, Che & Wang, 2022*). The process of VMD processing for wind speed data is shown in Fig. 4.

The VMD processing results of Example 1 and Example 2 are shown in Figs. 5 and 6, respectively. The original wind speed is decomposed into five components (intrinsic mode function (IMF) 1, IMF2, IMF3, IMF4, IMF5), each component represents a sub sequence of wind speeds with different frequencies but relatively stable.

In this article, the range standardization method is employed to normalize the original wind speed data. This method involves converting the sample data values into a range between 0 and 1. By applying this normalization technique, the wind speed data is standardized, enabling easier comparison and analysis across different scales or datasets (*Liu et al., 2018*). The specific calculation method is shown in Eq. (1):

$$X_{nor} = \frac{x - x_{\min}}{x_{\max} - x_{\min}} \tag{1}$$

where $x$ represents the true value of the data; $x_{\max}$ depicts the maximum value of the data; $x_{\min}$ denotes the minimum value of the data; $X_{nor}$ stands for the normalized value obtained after applying the range standardization method.

In order to better compare the predicted value of wind speed with the real value, it is necessary to perform inverse normalization on the predicted value, and the equation is as follows:

$$x = X_{nor}(x_{\max} - x_{\min}) + x_{\min}. \tag{2}$$

# BASIC PRINCIPLES OF LSTM, BILSTM AND TSO

## Basic principles of LSTM

On the basis of recurrent neural networks (RNN), LSTM introduces several components such as forget gate, input gate, memory unit update and output gate to enhance the performance of the repetitive hidden layer neuron structure. These additions enable

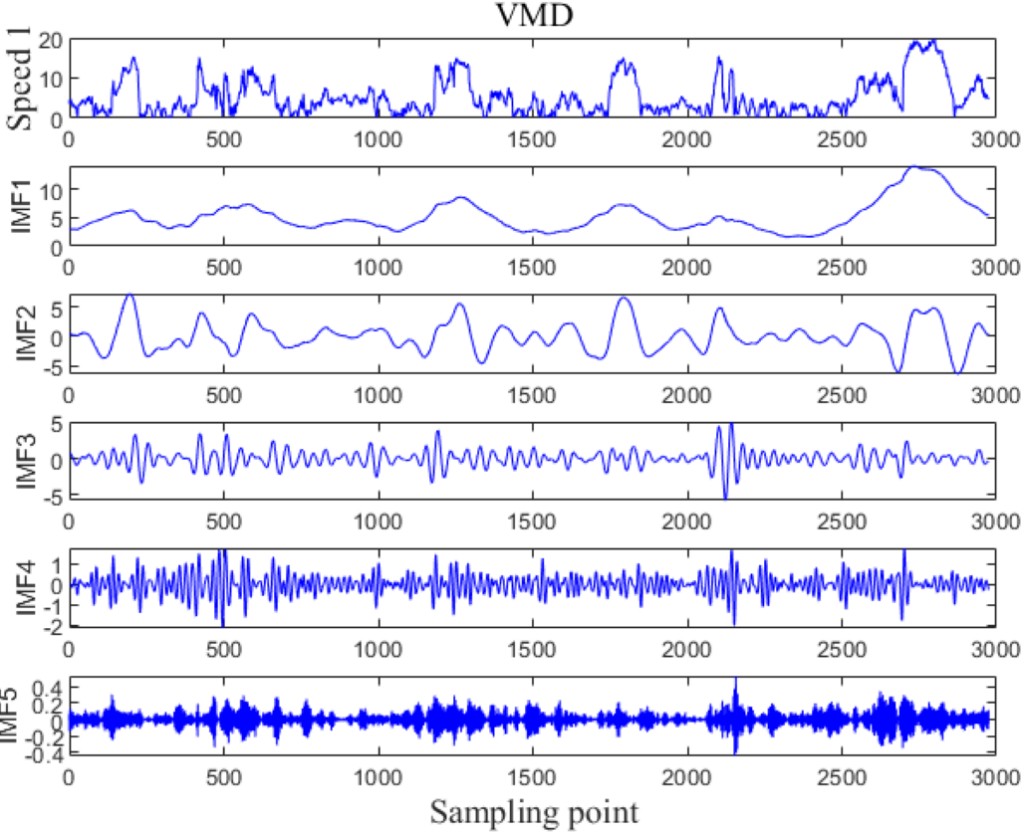

**Figure 5** VMD processing results for Example 1.

LSTM to effectively store and remain historical messages and control the convergence of gradients during training (*Fukuoka et al., 2018*). The issue of gradient vanishing and gradient exploding during RNN training is effectively addressed and resolved (*Germánico & Pablo, 2022*). The basic structure of LSTM hidden layer neurons is shown in Fig. 7, where $x_t$ refers to the input data, $h_t$ represents the output of the recurrent layer, $c_t$ denotes the output of the memory unit, $h_{t-1}$ signifies the output of the previous recurrent layer, and $c_{t-1}$ represents the output of the previous memory unit.

The function of the forget gate is to determine whether to forget information, usually with a certain probability to control whether to forget the corresponding information. Use the sigmoid function to process $h_{t-1}$ and $x_t$, in order to obtain the output $f_t$. Since the value range of $f_t$ is [0, 1], $f_t$ also represents the probability of forgetting, and its expression is:

$$f_t = \sigma(W_f h_{t-1} + U_f x_t + b_f) \tag{3}$$

where, $\sigma$ refers to the activation function, $W_f$ and $U_f$ represent the coefficients of the linear relationship, $b_f$ is the bias coefficient.

The input gate is mainly accountable for dealing with the input of the current sequence position, consisting of two components: the output of the first part $i_t$ with the matching

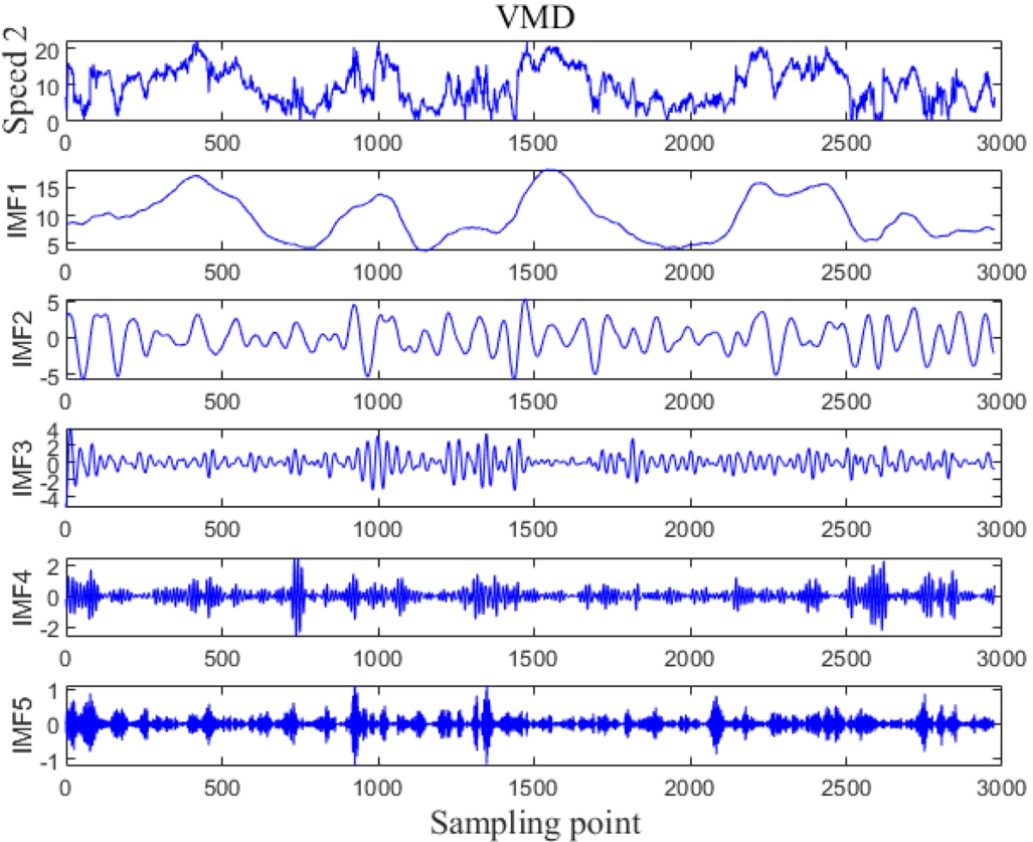

**Figure 6 VMD processing results for Example 2.**

activation function $\sigma$, and the output of the second part $a_t$ with the activation function *tanh*. The expression for the input gate can be represented as follows:

$$i_t = \sigma(W_i h_{t-1} + U_i x_t + b_i) \tag{4}$$

$$a_t = tanh(W_a h_{t-1} + U_a x_t + b_a). \tag{5}$$

$W_i$, $U_i$, $W_a$ and $U_a$ represent the coefficients of linear relationship; $b_i$ and $b_a$ stand for the bias coefficients of the linear relationship.

The cell state $c_t$ is affected by the outputs of the forget gate and the input gate. As a result, $c_t$ can be separated into two components: the first component refers to the multiplication of $c_{t-1}$ and $f_t$, and the second component involves the product of $i_t$ and $a_t$. $c_t$ can be obtained by adding the two products. For the specific calculation process, see Eq. (6):

$$c_t = c_{t-1} \otimes f_t + i_t \otimes a_t. \tag{6}$$

The output gate is accountable for determining the final output information, and the update of $h_t$ is also divided into two components. The first component is $o_t$, which is

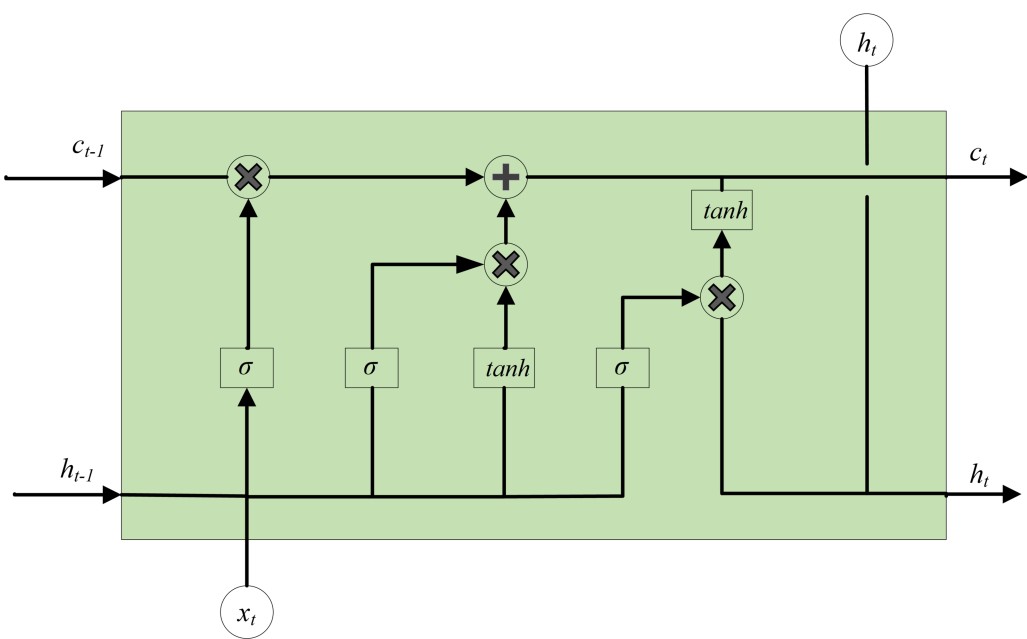

**Figure 7** **Basic structure of LSTM hidden layer neurons.**

composed of $h_{t-1}$, $x_t$ and activation function $\sigma$. The second component consists of the hidden state $c_t$ and the activation function $tanh$.

$$o_t = \sigma(W_o h_{t-1} + U_o x_t + b_o) \tag{7}$$

$$h_t = o_t \otimes tanh(c_t) \tag{8}$$

where $W_o$ and $U_o$ are also the coefficients of linear relationship, and $b_o$ is the bias coefficient.

Through these four mechanisms, LSTM can selectively control the flow of information, giving memory cells the ability to preserve long-term information dependencies, while also preventing internal gradients from being disturbed by external factors during the training process (*Wang et al., 2023*).

## Basic principles of BiLSTM

BiLSTM is based on classical LSTM by using two layers of independent cell units, namely forward cell unit and reverse cell unit (*He et al., 2023*). The model structure is shown in Fig. 8. In Fig. 8, $x_{t-1}$, $x_t$ and $x_{t+1}$ represent the inputs at *t-1*, *t* and *t+1*, respectively. $h_{t-1}$, $h_t$ and $h_{t+1}$ denote the outputs at *t-1*, *t* and *t+1* in forward and reverse states, respectively. $y_{t-1}$, $y_t$ and $y_{t+1}$ stand for the final outputs at *t-1*, *t* and *t+1*. In the forward state, input $x_{t-1}$, $x_t$ and $x_{t+1}$ to get the forward outputs $h_{t-1}$, $h_t$ and $h_{t+1}$; In the reverse state, input $x_{t+1}$, $x_t$ and $x_{t-1}$ to obtain the reverse outputs $h_{t+1}$, $h_t$ and $h_{t-1}$. Finally, the two sets of

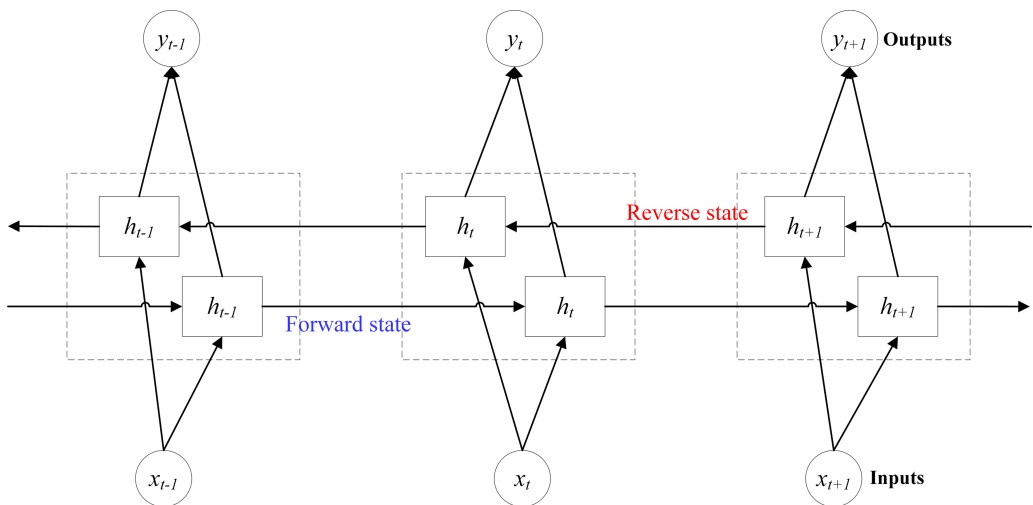

**Figure 8** Model structure diagram of BiLSTM.

outputs are combined, and finally the outputs $y_{t-1}$, $y_t$ and $y_{t+1}$ corresponding to the inputs $x_{t-1}$, $x_t$ and $x_{t+1}$ are obtained (*Gao & Hao, 2023*).

The processing method of BiLSTM is the same as that of classical LSTM, but BiLSTM traverses the input data twice in forward and reverse ways, increasing the training times, thus improving the wind speed prediction accuracy under micrometeorological parameters (*Liu et al., 2022*), and the final prediction result is more accurate than that of a single LSTM (*Zheng, Zhou & Nan, 2023*).

## Basic principles of TSO

TSO is a new type of intelligent optimization algorithm, which mimics the foraging behavior of tuna populations to solve the optimization problems. It has simple structure, few adjustment parameters, fast convergence speed, strong optimization capability and easy implementation (*Guo et al., 2022*)

Similar to other intelligent optimization algorithms, tuna populations are typically randomly initialized within the search space, and the initialization process can be represented as:

$$X_i^{\text{int}} = rand \cdot (ub - lb) + lb, i = 1, 2, \cdots, np \tag{9}$$

$X_i^{\text{int}}$ represents the initial position of the *i-th* individual, *ub* and *lb* denote the upper and lower bounds of the search space, respectively. *np* stands for the number of tuna populations, and *rand* represents a random vector uniformly distributed within the range [0, 1].

Tuna populations usually choose a spiral foraging method. During foraging, tuna individuals exchange prey information with each other. Due to each individual following the previous individual to forage, tuna populations can share hunting information in real-time (*Liu, Fan & Li, 2023*). Equations (10) and (11) represent the mathematical model of the spiral foraging strategy.

$$X_i^{t+1} = \begin{cases} \alpha_1 \cdot \left(X_{best}^t + \beta \cdot \left|X_{best}^t - X_i^t\right|\right) + \alpha_2 \cdot X_i^t, i=1 \\ \alpha_1 \cdot \left(X_{best}^t + \beta \cdot \left|X_{best}^t - X_i^t\right|\right) + \alpha_2 \cdot X_{i-1}^t, i=2,3,\cdots,np \end{cases} \tag{10}$$

$$\beta = e^{bl} \cdot cos(2\pi b) \tag{11}$$

$$l = e^{3\cos(((t_{max}+1/t)-1)\pi)} \tag{12}$$

$X_i^{t+1}$ represents the *i-th* individual at the *t+1* iteration, $X_i^t$ represents the *i-th* individual at the *t* iteration, $X_{best}^t$ stands for the current best individual, $\alpha 1$ and $\alpha 2$ indicate the weight coefficients that control the movement trend of an individual towards to the best individual and the previous individual, respectively. $t$ represents the number of current iterations, $t_{max}$ stands for the maximum iteration number, and $b$ represents a random number evenly distributed between 0 and 1.

When the optimal individual of tuna populations is unable to find food, blindly following it is not conducive to group foraging. Thus, it becomes necessary to generate a random coordinate within the search space as a reference for the search of tuna populations. This enables each individual to explore in a broader area and gives TSO the capability to explore globally (*Xue, Liu & Wang, 2022*). The specific mathematical model description can be found in Eq. (13):

$$X_i^{t+1} = \begin{cases} \alpha_1 \cdot \left(X_{rand}^t + \beta \cdot \left|X_{rand}^t - X_i^t\right|\right) + \alpha_2 \cdot X_i^t, i=1 \\ \alpha_1 \cdot \left(X_{rand}^t + \beta \cdot \left|X_{rand}^t - X_i^t\right|\right) + \alpha_2 \cdot X_{i-1}^t, i=2,3,\cdots,np \end{cases} \tag{13}$$

where $X_{rand}^t$ indicates a randomly generated reference point within the search space. Such heuristic algorithm conducts extensive global exploration in the early stage, and gradually transitions to more precise local development. Therefore, with the increasing number of iterations of the algorithm, the reference object of the tuna populations is transformed from random coordinates to the optimal individual.

Apart from the spiral foraging strategy, tuna populations also adopt another foraging strategy, called parabolic foraging strategy, which is to form a parabolic shape with prey as a reference point, and then gradually narrow the search area (*Xie, Han & Zhou, 2021*). The probability of the tuna population choosing the two foraging strategies is basically the same, both are 50%. The mathematical model description of parabolic foraging strategy is shown by Eqs. (14) and (15):

$$X_i^{t+1} = \begin{cases} X_{best}^t + rand \cdot (X_{best}^t - X_i^t) + TF \cdot p^2 \cdot (X_{best}^t - X_i^t), rand < 0.5 \\ TF \cdot p^2 \cdot X_i^t, rand \geq 0.5 \end{cases} \tag{14}$$

$$p = (1 - \frac{t}{t_{max}})^{\frac{t}{t_{max}}} \tag{15}$$

$TF$ represents a random number within a value range of $[-1, 1]$.

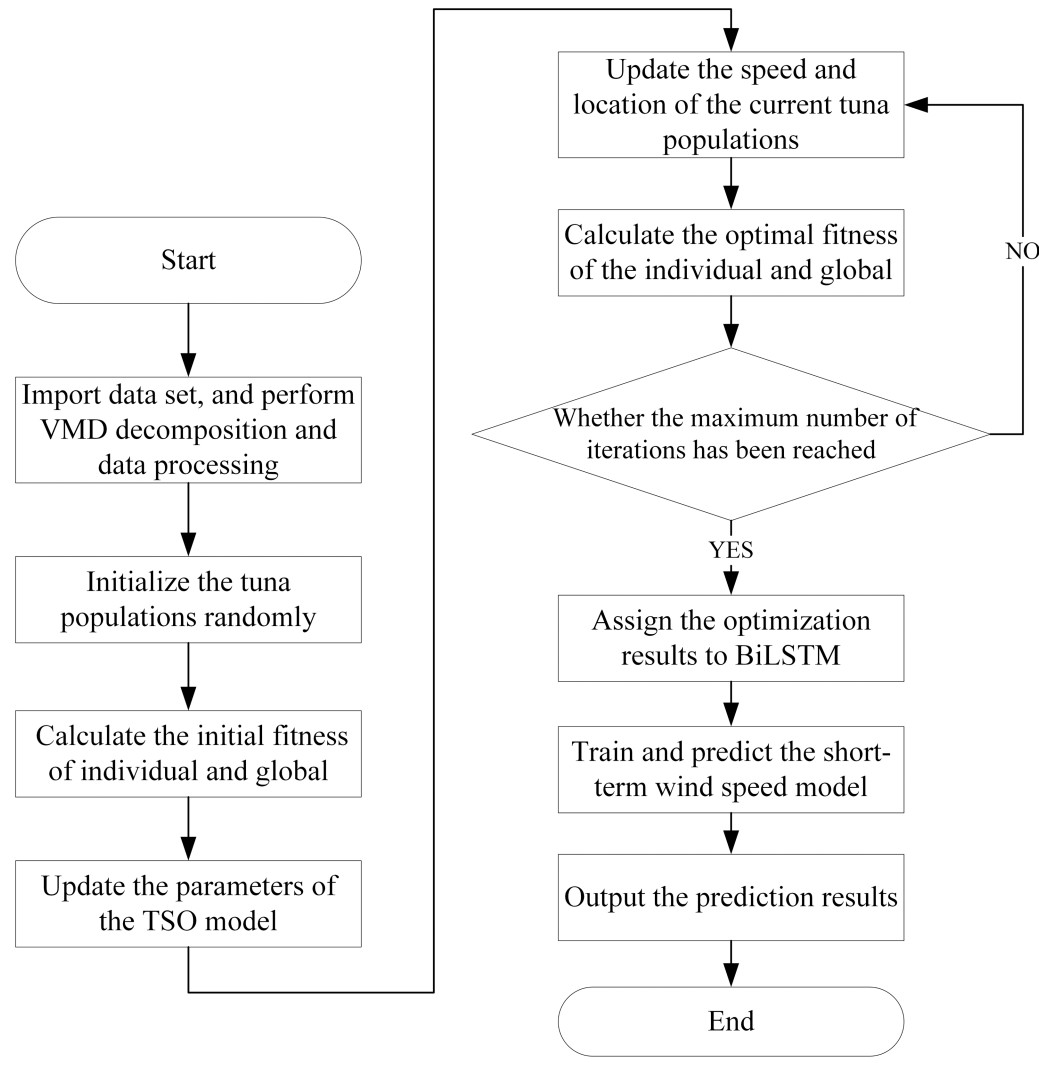

**Figure 9** **SWSP flowchart based on TSO-VMD-BiLSTM.**

## RESULTS

According to the basic principles of TSO-VMD-BiLSTM combination method, this article draws the implementation flow chart of SWSP model based on this method, as shown in Fig. 9.

Step 1: Import wind speed data set, and perform VMD decomposition and data processing.

Step 2: Initialize the tuna populations randomly.

Step 3: Record the location of the current tuna populations and calculate the initial fitness of individual and global.

Step 4: Update the parameters of the TSO, and update the speed and location of the current tuna populations, then calculate the optimal fitness of the individual and global.

**Table 1  Model parameters of various algorithms.**

| Algorithm | Parameter | |
|---|---|---|
| VMD | $\alpha = 2500$ | $K = 5$ |
| | DC = 0 | tol = 1e−7 |
| TSO | $np = 8$ | $t_{max} = 8$ |
| | $\alpha_1 = 0.8$ | $\alpha_2 = 0.2$ |
| LSTM | Num_units = 50 | Max_iteration = 300 |
| | Ilr = 0.005 | Lrdp = 200 |
| | Lr $df = 0.5$ | |

Step 5: Check if the maximum number of iterations has been reached. If not, go back to the step 4. Otherwise, record the location information and optimal parameters of the current tuna populations.

Step 6: Assign the optimization results to BiLSTM.

Step 7: Train and predict the short-term wind speed model according to the results of the previous steps.

Step 8: Carry on the inverse normalization process on the prediction results and output them.

The model parameters of various algorithms adopted in this article are displayed in Table 1.

With a view to fully validate the feasibility and effectiveness of the TSO-VMD-BiLSTM prediction model, this article conducts a comparative study with three models: LSTM, VMD-LSTM and TSO-VMD-LSTM. The evaluation indicators used include absolute error (AE), root mean squared error (RMSE) and mean absolute error (MAE), mean absolute percentage error (MAPE), and specific expressions of evaluation indicators are shown in Eqs. (16)–(19).

$$AE = \left| \overline{y}i - yi \right| \tag{16}$$

$$RMSE = \sqrt{\frac{1}{n}\sum_{i=1}^{n}(\overline{y}i - yi)^2} \tag{17}$$

$$MAE = \frac{1}{n}\sum_{i=1}^{n}\left| \overline{y}i - yi \right| \tag{18}$$

$$MAPE = \frac{1}{n}\sum_{i=1}^{n}\left| \frac{\overline{y}i - yi}{yi} \right| \times 100\% \tag{19}$$

where $yi$ denotes the true value, $\overline{y}i$ indicates the predicted value, $n$ stands for the number of samples in the test set.

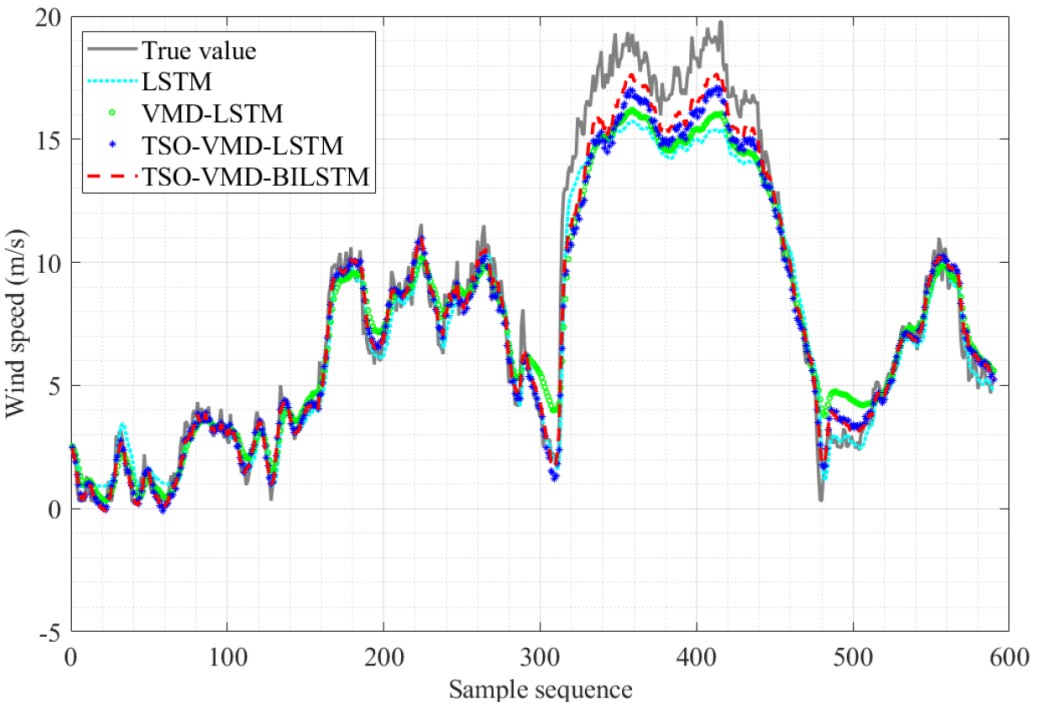

**Figure 10** Prediction results of the four models in Example 1.

The verification environment used in this article includes Windows 10 operating system, Intel Core i7 processor, 4 GB stand-alone graphics card, 16 GB memory environment, and Matlab 2022 software.

## Example 1

The results of the four SWSP models are shown in Fig. 10. Obviously, the predicted value curve of the TSO-VMD-BILSTM model exhibits a tendency to closely align with the true value curve, especially in the high wind speed range, that is, the sample sequence 310–480, and its prediction effect is significantly better than that of the other three models.

Take the true value of wind speed as the horizontal axis and the predicted value of the four models as the vertical axis, and observe the linearity of the curve (*Liu, Yu & Cang, 2015*). The linearity is directly proportional to the prediction accuracy, that is, the better linearity, the higher prediction accuracy. The linearity curves of the four models in Example 1 are shown in Fig. 11, from which two pieces of information can be obtained. (1) Because of the randomness and intermittency of wind speed, the linearity curves show an unordered state; (2) In the unordered state, the more concentrated the linearity curve distribution of the prediction model, the higher the accuracy. Obviously, compared to the other three models, the predictive curve of the TSO-VMD-BiLSTM model has a more concentrated and compact linear distribution, which means that the model has the best predictive performance.

Figure 12 shows the AE values of the four prediction models in Example 1. The fluctuation range of AE values of TSO-VMD-BiLSTM is about [0, 2.52]. Except for the

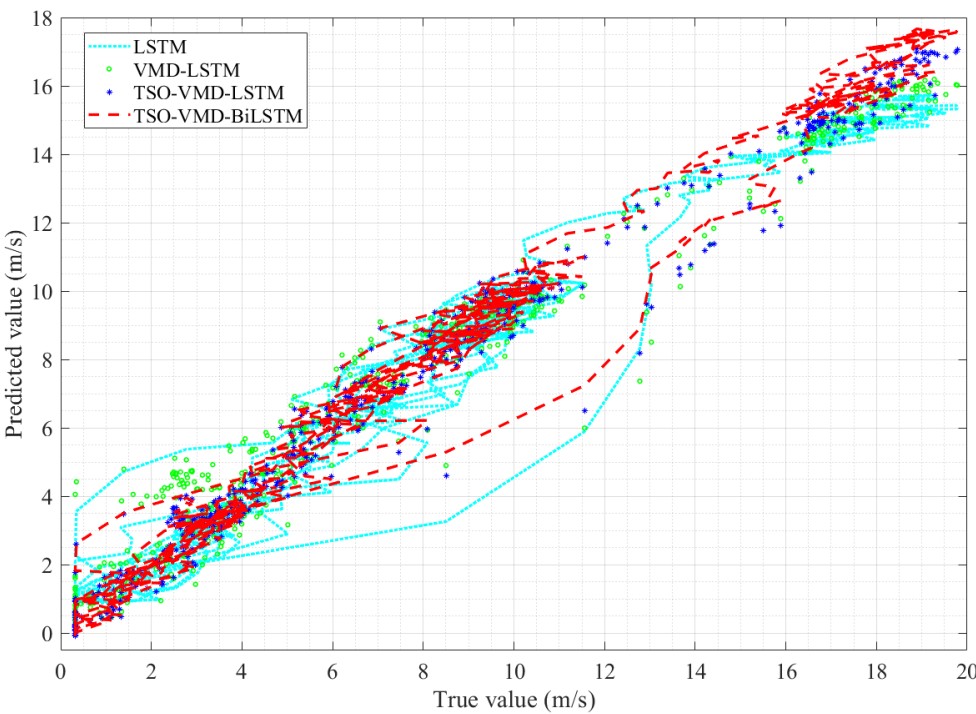

**Figure 11  Linearity curves of the four models in Example 1.**

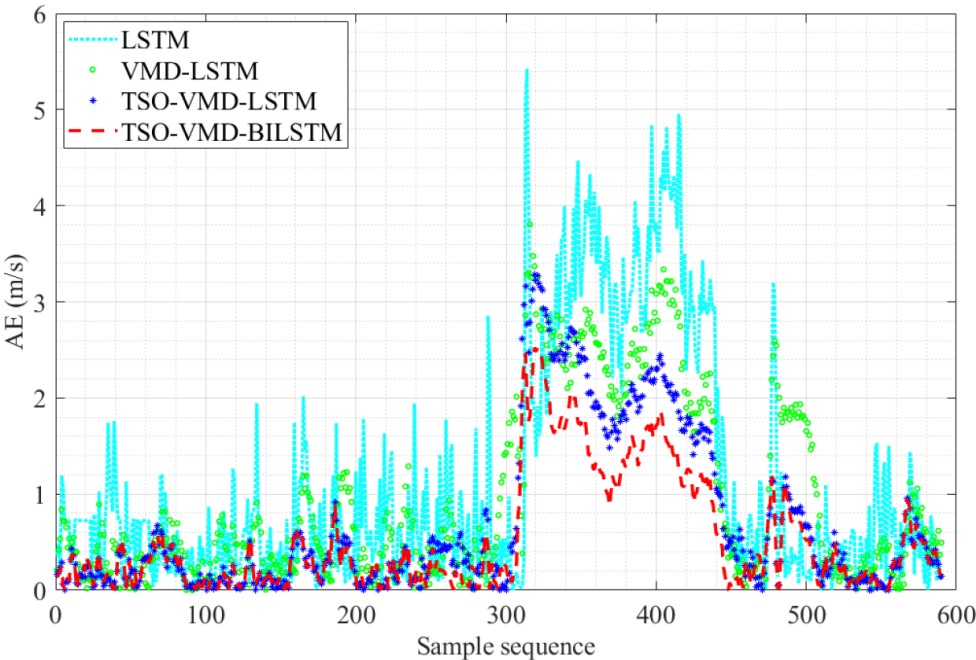

**Figure 12  AE diagram of the four prediction models in Example 1.**

**Table 2   Results of other evaluation indicators of the four prediction models in Example 1.**

| Model | Evaluation indicator | | |
| --- | --- | --- | --- |
| | RMSE | MAE | MAPE |
| LSTM | 1.69 | 1.15 | 15.27% |
| VMD–LSTM | 1.42 | 0.98 | 12.53% |
| TSO-VMD-LSTM | 1.09 | 0.70 | 9.50% |
| TSO-VMD-BILSTM | 0.81 | 0.54 | 6.89% |

high wind speed range, the AE values of the vast majority of test samples are less than 1 m/s, and the AE values of each test sample are lower than the other three models in the high wind speed range. Compared to the TSO-VMD-LSTM model, the overall fluctuation range of TSO-VMD-BiLSTM has been reduced, and the prediction results are more accurate than unidirectional LSTM. This indicates that the introduction of BiLSTM effectively captures micro meteorological parameters.

The other evaluation indicators of the four prediction models in Example 1 are shown in Table 2. The RMSE value of TSO-VMD-BiLSTM is only 0.81, which is 0.88, 0.61 and 0.28 lower than that of LSTM, VMD-LSTM and TSO-VMD-LSTM, respectively. MAE is only 0.54, which is 0.61 less than LSTM, 0.44 less than VMD-LSTM, and 0.16 less than TSO-VMD-LSTM. MAPE is only 6.89%, which is 8.38%, 5.64% and 2.61% lower than LSTM, VMD-LSTM and TSO-VMD-LSTM, respectively. Therefore, by integrating various evaluation indicators, it can be found that the SWSP model based on TSO-VMD-BiLSTM has the best performance and the highest accuracy.

## Example 2

The SWSP results of the four prediction models in Example 2 are shown in Fig. 13. It is not difficult to see that TSO-VMD-BiLSTM exhibits good tracking performance, and its predicted value curve demonstrates a closer resemblance to the true value curve compared to the other three prediction models. In addition, the predicted values closely track and reflect the changes observed in the true values.

The linearity curves of the four prediction models in Example 2 are shown in Fig. 14, and we can get the same conclusion as in Example 1. The linearity curve of TSO-VMD-BiLSTM is the most concentrated and tight distribution, followed by TSO-VMD-LSTM and VMD-LSTM, and finally LSTM. Therefore, the TSO-VMD-BiLSTM model has the highest prediction accuracy.

Figure 15 shows the AE graph of the TSO-VMD-BiLSTM prediction model. By comparing the AE values of each model, it is noticeable that the TSO-VMD-BiLSTM model has the best prediction performance, the smallest AE value, and the error value is controlled within 2 m/s. Moreover, the AE values of most test samples are less than 1 m/s. Similarly, compared with TSO-VMD-LSTM, the overall error fluctuation range has been decreased, especially at sample sequence 240, BiLSTM still shows good ability to capture micro meteorological parameters.

Table 3 illustrates the results of other evaluation indicators of the four prediction models in Example 2. The RMSE of TSO-VMD-BiLSTM is only 0.48, which is 1.07,

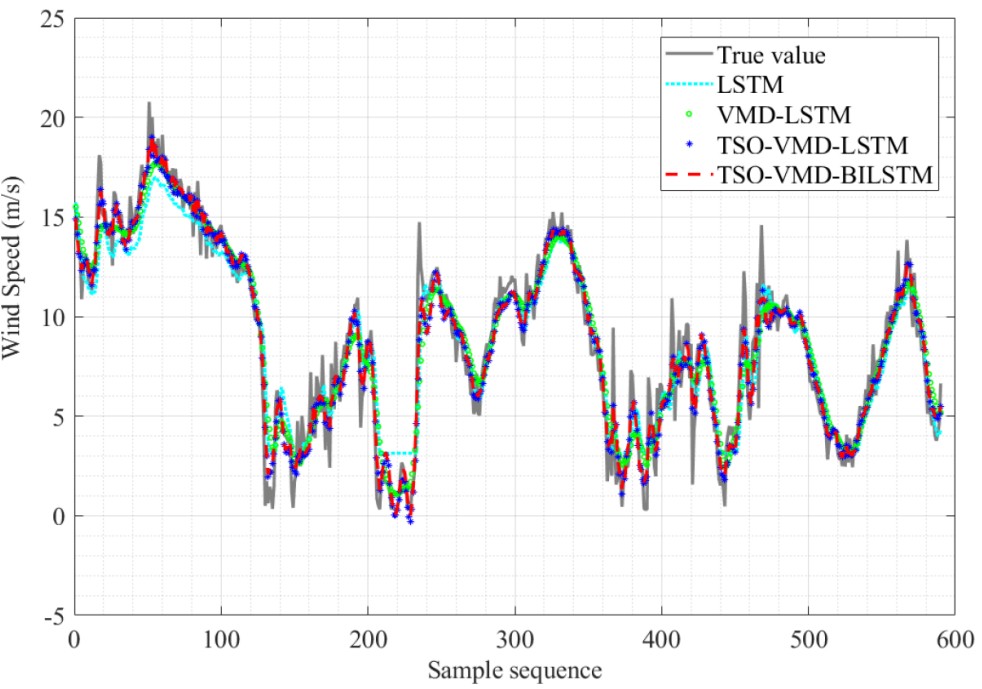

**Figure 13 Prediction results of the four models in Example 2.**

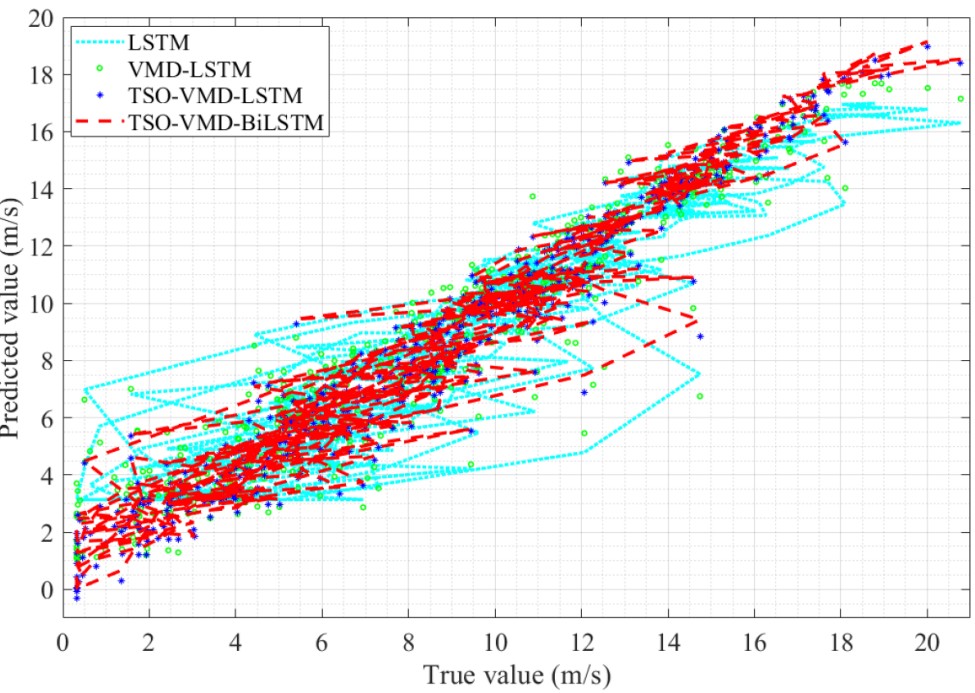

**Figure 14 Linearity curves of the four models in Example 2.**

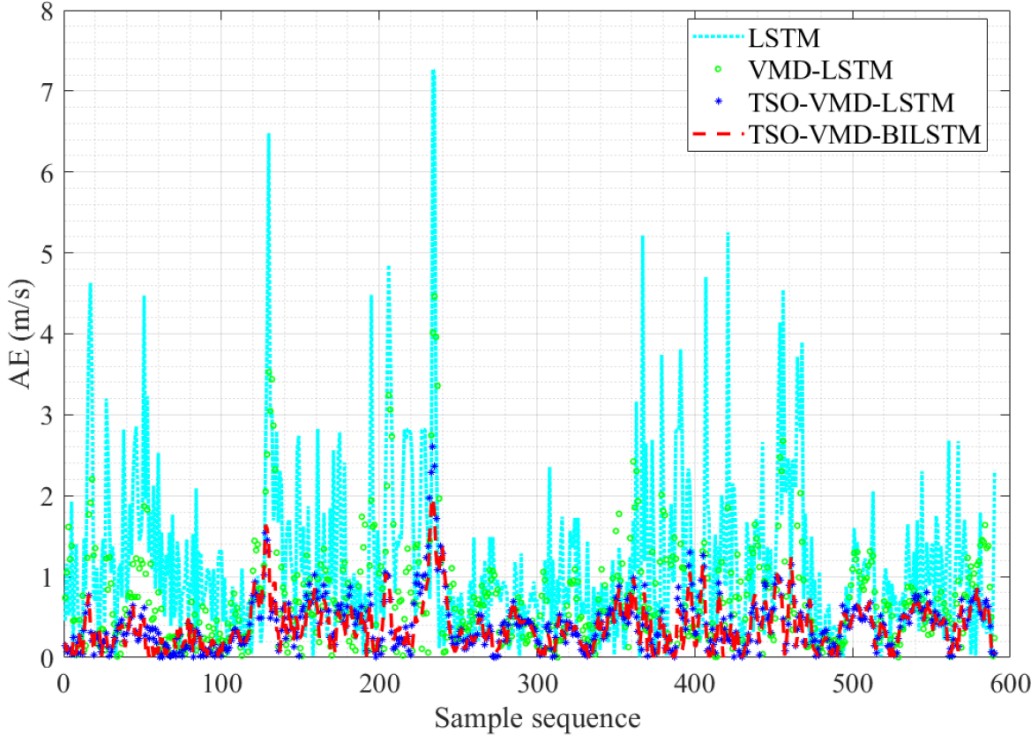

**Figure 15  AE diagram of the four prediction models in Example 2.**

**Table 3  Results of other evaluation indicators of the four prediction models in Example 2.**

| Model | Evaluation indicator | | |
|---|---|---|---|
| | RMSE | MAE | MAPE |
| LSTM | 1.55 | 1.17 | 8.54% |
| VMD-LSTM | 0.95 | 0.74 | 5.98% |
| TSO-VMD-LSTM | 0.53 | 0.41 | 3.34% |
| TSO-VMD-BILSTM | 0.48 | 0.38 | 3.17% |

0.47 and 0.05 lower than LSTM, VMD-LSTM and TSO-VMD-LSTM, respectively. The MAE is only 0.38, which is 0.79 less than LSTM, 0.36 less than VMD-LSTM, and 0.03 less than TSO-VMD-LSTM, respectively. MAPE is only 5.98%, which is 5.37%, 2.81% and 0.17% better than LSTM, VMD-LSTM and TSO-VMD-LSTM, respectively. All in all, TSO-VMD-BiLSTM prediction model has the highest accuracy and the best performance.

## CONCLUSIONS

In this article, a SWSP model based on TSO-VMD-BiLSTM is proposed. VMD is employed to extract features from wind speed data, and then a BiLSTM-based prediction model is developed, with TSO optimizing the model parameters. Selecting the wind speed data of a certain wind farm in the valley and peak months of 2022 for example verification,

and comparing it with wind speed prediction models based on LSTM, VMD-LSTM, and TSO-VMD-LSTM, leads to the conclusions that follow:

(1) Comparing the validation results of LSTM and VMD-LSTM prediction models, the effectiveness of the VMD algorithm in extracting significant wind speed characteristics, as evidenced by improvements across all evaluation metrics including AE, RMSE, MAE, and MAPE.

(2) The assessment of VMD-LSTM and TSO-VMD-LSTM models demonstrates the parameter optimization effect of TSO algorithm is significant, accompanied by a decrease in the data values of evaluation indicators.

(3) Comparing the validation results of TSO-VMD-LSTM and TSO-VMD-BiLSTM prediction models, it illustrates the superior ability of the BiLSTM algorithm to capture micrometeorological parameters, achieving the highest accuracy among the compared models.

### Funding
The authors received no funding for this work.

### Competing Interests
The authors declare there are no competing interests.

### Author Contributions
- Qi Wang conceived and designed the experiments, performed the experiments, analyzed the data, prepared figures and/or tables, authored or reviewed drafts of the article, and approved the final draft.
- Lei Zhang conceived and designed the experiments, performed the experiments, performed the computation work, prepared figures and/or tables, and approved the final draft.

### Data Availability
The raw data and code are available in the Supplementary Files.

### Supplemental Information
Supplemental information for this article can be found online at http://dx.doi.org/10.7717/peerj-cs.2032#supplemental-information.

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
