# Peer review of "Short-term wind speed prediction of wind farm based on TSO-VMD-BiLSTM"

_PeerJ Computer Science, doi:10.7717/peerj-cs.2032_

## Round 0.1 · original submission · Major Revisions

This manuscript has been carefully examined by three experts in this field. The reviewers recognised some interesting findings and results but also raised reasonable comments and suggestions. Therefore, I would like the authors to carefully consider the reviewers' comments and make point-by-point revisions to improve the manuscript.

**Language Note:** The review process has identified that the English language must be improved. PeerJ can provide language editing services - please contact us at [email protected] for pricing (be sure to provide your manuscript number and title). Alternatively, you should make your own arrangements to improve the language quality and provide details in your response letter. – PeerJ Staff

Reviewer 1 ·

Basic reporting

(1) When summarizing short-term wind speed prediction methods, the authors should attempt to supplement some of the latest research findings.
(2) The correlation between a few references and the research content of the paper is not too high, so it is recommended to replace them.

Experimental design

(1) In the process of sample data processing, the paper needs to clarify the division ratio between the training set and the test set, which is crucial for controlling prediction accuracy.
(2) The short IMF in the paper does not give the full name, please complete it.
(3) The authors proposed a SWSP method based on TSO-VMD-BiLSTM, which is obviously a combination method. Does the combination of the three algorithms have a certain logical basis?
(4) Does BiLSTM run in such a way that two LSTMs work in parallel? Or are there other ways of working? Authors need to explain.
(5) What is the probability that a tuna population will choose one of the two foraging strategies?

Validity of the findings

(1) The linearity curve of the prediction model in this paper is complicated, and how to analyze the linearity curve needs to be strengthened.
(2) In this paper, two examples are used to verify the SWSP based on TSO-VMD-BiLSTM, but the paper does not explain the relevant conditions of verification clearly, please complete.
(3) Can the two example studies in the paper reach the same conclusion? If possible, please describe it in the appropriate place in the paper.

Additional comments

no comment

Cite this review as

·

Basic reporting

1. L18 Besides meteorological factors, what other factors can affect short-term wind speed prediction?
2. L39-40 Although it is important for prediction methods to accurately predict short-term wind speed, these methods are data-driven. Please explain whether wind speed data has an impact on prediction accuracy.
3. L60 The authors propose a combined short-term wind speed prediction method based on TSO-VMD-BiLSTM, and what are the advantages of this method?

Experimental design

1. L94 What is the purpose of the authors choosing two examples to study? What are the benefits of evaluating prediction accuracy?
2. L108 What are the advantages of VMD algorithm for wind speed data processing? The authors need to complete it.
3. L188 BiLSTM has an additional reverse process compared with LSTM. Will this increase the computational cost?
4. L205 There are many intelligent optimization methods, why did the authors choose the tuna swarm optimization algorithm? What are its advantages?

Validity of the findings

1. L283, L323 The authors have conducted a lot of simulation research in this paper, but the actual simulation environment needs to be explained clearly.
2. L275-278 Many errors are introduced in this paper as evaluation indicators. How are these errors classified? Or what kind of performance are they used to evaluate?

Additional comments

no comment

Reviewer 3 ·

Basic reporting

Section 1:
(1) For the overview of SWSP methods, it is recommended that the chronological order be described hierarchically.
Section 4:
(1) The flowchart in Figure 9 does not seem to be consistent with the steps it describes. Please give a specific reason or update the flowchart and its description.
All sections:
(1) Please improve the English language.

Experimental design

Section 2:
(1) In this paper, January and May are selected as the object of example study, and please give the specific reasons.
(2) When the VMD method is used to process the wind speed data, the original wind speed data is divided into 5 modes in the paper. How are these 5 modes determined? Is there a specific theoretical basis?
Section 3:
(1) BiLSTM algorithm is indeed more accurate than LSTM in many applications, but does it also bring some disadvantages? How do the authors weigh the advantages and disadvantages of the algorithm?
(2) TSO is a relatively new optimization algorithm in recent years, and introducing this method into SWSP has a certain degree of innovation. TSO generally has two foraging strategies, please fill them out.

Validity of the findings

Section 4:
(1) The linearity curve is used to analyze the deviation between the predicted value and the real value. What is its significance?

Additional comments

no comments

Cite this review as

---

## Round 0.2 · accepted · Accept

The authors have addressed all of the reviewers' comments. The manuscript is ready for publication.

Reviewer 1 ·

Basic reporting

I have no comment on this revised paper

Experimental design

I have no comment on this revised paper

Validity of the findings

I have no comment on this revised paper

Additional comments

I have no comment on this revised paper

Cite this review as

Reviewer 3 ·

Basic reporting

good

Experimental design

good

Validity of the findings

good

Cite this review as